# Difficult-to-treat resistant *Pseudomonas aeruginosa* infections in Lebanese hospitals: Impact on mortality and the role of initial antibiotic therapy

**Rania Itani** [1]*, **Hani M. J. Khojah**[2], **Tareq L. Mukattash**[3], **Patricia Shuhaiber**[4], **Hamza Raychouni**[5,6,7], **Carole Dib**[8], **Mariam Hassan**[9], **Abdalla El-Lakany**[10,11]

**1** Pharmacy Practice Department, Faculty of Pharmacy, Beirut Arab University, Beirut, Lebanon, **2** Department of Pharmacy Practice, College of Pharmacy, Taibah University, Madinah, Kingdom of Saudi Arabia, **3** Department of Clinical Pharmacy, Faculty of Pharmacy, Jordan University of Science and Technology, Irbid, Jordan, **4** Pharmacy Department, Mount Lebanon Hospital Balamand University Medical Center, Hazmieh, Lebanon, **5** Intensive Care Unit, Anesthesia Department, Central Military Hospital, Military Healthcare, Lebanese Army, Beirut, Lebanon, **6** Intensive Care Unit, Respiratory Care Department, American University of Beirut Medical Center, Beirut, Lebanon, **7** Intensive Care Unit, Aboujaoudé Hospital, Jal el Dib, Maten, Lebanon, **8** Pharmacy Department, Lebanese Hospital Geitaoui University Medical Center, Beirut, Lebanon, **9** Emergency Department, Sahel General Hospital, Beirut, Lebanon, **10** Department of Pharmaceutical Sciences, Faculty of Pharmacy, Beirut Arab University, Beirut, Lebanon, **11** Department of Pharmacognosy, Faculty of Pharmacy, Alexandria University, Alexandria, Egypt

* r.itani@bau.edu.lb

## Abstract

### Background

Difficult-to-treat resistant (DTR) *Pseudomonas aeruginosa* infections have emerged as a significant global public health threat, characterized by limited treatment options and a heightened mortality risk. This study aimed to assess the appropriateness of initial antibiotic therapy, estimate 30-day all-cause mortality, and determine the impact of DTR *P. aeruginosa* infections on mortality.

### Methods

A retrospective, multicenter study was conducted at four teaching hospitals in Beirut, Lebanon, between January 2021 and December 2023. The primary outcome was 30-day all-cause mortality. Kaplan-Meier survival analysis was used to assess time-to-mortality, and the log-rank test was applied to compare survival outcomes relative to DTR infections and the appropriateness of initial antibiotic therapy. Multivariable logistic regression was performed to identify predictors of mortality.

### Results

Out of 2,639 screened cases, 477 patients met the inclusion criteria. Respiratory tract infections accounted for 38.8% of cases. Carbapenem-resistant *P. aeruginosa*

**Data availability statement:** The datasets generated and analyzed during this study are not publicly available due to restrictions imposed by the IRBs and ethical committees of the enrolled hospitals, which are intended to protect patient privacy and confidentiality. The data contain potentially identifying or sensitive patient information, which could risk patient re-identification. Additionally, given that this study assessed hospital management practices, further confidentiality concerns apply. In accordance with ethical standards, access to these datasets requires a formal request to the corresponding author (r.itani@bau.edu.lb) or to the IRB at Beirut Arab University (irb@bau.edu.lb), along with approval from the hospitals' IRBs and ethical committees. All relevant data supporting the conclusions of this study are included within the article. Furthermore, the data will be securely stored in Beirut Arab University's institutional repository, ensuring long-term availability and accessibility through the IRB.

**Funding:** The author(s) received no specific funding for this work.

**Competing interests:** The authors have declared that no competing interests exist.

(CRPA) comprised nearly one-third of isolates, and 15.3% were categorized as DTR. The most common empirical antibiotics were piperacillin-tazobactam (33.9%) and meropenem (24.5%). Inappropriate initial antibiotic therapy was observed in 43.8% of cases, with 33.8% of patients receiving antibiotics to which the pathogen was resistant. DTR infections were significantly more likely to be associated with inappropriate therapy (odds ratio [OR] = 4.21, 95% CI = 2.43–7.32, P < 0.001). The 30-day all-cause mortality rate was 14.8%, with a mean time-to-mortality of 13.29 ± 9.81 days. Patients who received inappropriate therapy had a shorter time-to-mortality (11.76 ± 8.80 days) compared to those receiving appropriate therapy (15.46 ± 10.90 days, P = 0.03). Predictors of mortality included DTR *P. aeruginosa* infection (adjusted odds ratio [AOR] = 2.48, 95% CI = 1.32–4.63, P < 0.01), and inappropriate initial therapy (AOR = 1.40, 95% CI = 1.04–2.35, P < 0.01).

## Conclusion

DTR *P. aeruginosa* infections and inappropriate initial antibiotic therapy are associated with increased mortality risk in hospitalized patients.

---

## 1. Background

The rise of antimicrobial resistance (AMR) has emerged as a critical global public health threat, jeopardizing the effectiveness of existing therapeutic options. Among the foremost opportunistic pathogens contributing to AMR is *Pseudomonas aeruginosa*, a Gram-negative bacterium responsible for causing severe infections with high mortality rates [1]. In-hospital mortality among patients infected with *P. aeruginosa* ranges from 10.5% to 29.6% [2–6]. The bacterium's intrinsic ability to develop resistance, combined with complex adaptive mechanisms, renders *P. aeruginosa* particularly difficult to treat. As a member of the ESKAPE group of pathogens, it has been designated a critical priority by the World Health Organization, underscoring the urgent need for innovative research and the development of novel antibiotics [7].

Difficult-to-treat resistance (DTR) is a recently introduced classification that has become a valuable tool for both epidemiological research and clinical prognostication, aiding in the prediction of patient outcomes and mortality in the context of resistant infections [8]. DTR *P. aeruginosa* isolates exhibit resistance to all first-line antipseudomonal agents, including β-lactams and fluoroquinolones. As a result, treating DTR *P. aeruginosa* infections often requires the use of less effective and more toxic alternatives such as colistin, tigecycline, and aminoglycosides [8]. These infections are associated with poor clinical outcomes and significantly higher mortality rates. Globally, the prevalence of DTR *P. aeruginosa* ranged from 6% to 14.5% between 2018 and 2022 [9].

Despite growing concern over DTR *P. aeruginosa* due to its high risk of treatment failure and significant impact on patient outcomes, there is a paucity of well-designed studies investigating the impact of DTR *P. aeruginosa* infections on mortality. This gap is particularly pronounced in Lebanon, a lower-middle-income

country with a fragile healthcare infrastructure, which lacks national surveillance data and robust epidemiological tracking. Lebanon's challenges in controlling AMR arise from underdeveloped informatics systems, weak health systems governance, limited resources, poorly implemented antimicrobial stewardship (AMS) programs, and an absence of governmental oversight regarding antibiotic dispensing practices [10–13].

Lebanon has undertaken several initiatives at the national level to address AMR. In alignment with global surveillance efforts, the country joined the World Health Organization's (WHO) Global Antimicrobial Resistance Surveillance System (GLASS) to facilitate AMR data reporting and integrate laboratory data into international systems [14,15]. However, Lebanon has yet to submit AMR data to GLASS, as it is still in the process of building a national AMR surveillance system [16]. In 2019, the Ministry of Public Health (MoPH) established a National Antibiotic Resistance Committee and an AMR Task Force, which drafted a National Action Plan (NAP) to enhance AMR surveillance. The plan aimed to map laboratories for microbiologically reliable data, issue periodic national AMR reports, promote rational antimicrobial use, establish AMS programs in hospitals, develop national treatment guidelines for infectious diseases, and strengthen infection prevention and control measures [17].

Despite these efforts, the AMR-NAP was not officially integrated into the national health plan due to ongoing political instability, economic crises, and deficiencies in essential infrastructure, such as electricity and waste management. The lack of dedicated funding and financial investment further hindered its implementation. Consequently, national AMR surveillance data remain scarce, primarily confined to retrospective, single-center, or multi-center studies conducted in tertiary care settings. Available data are fragmented and lack epidemiological representativeness, limiting their utility in guiding policymakers and effectively allocating resources to mitigate antimicrobial resistance [16,18].

This study aimed to assess the appropriateness of initial antibiotic therapy administered to hospitalized patients with *P. aeruginosa* infections and estimate 30-day all-cause in-hospital mortality. Additionally, it sought to identify predictors of mortality, with a particular focus on the impact of DTR *P. aeruginosa* infections and antibiotic therapy on patient outcomes.

## 2. Methods

### 2.1. *Study design, setting, and institutional characteristics*

This retrospective, multicenter study was conducted across four large teaching hospitals in Beirut, Lebanon. It included patients hospitalized with *P. aeruginosa* infections over a three-year period, from January 1, 2021, to December 31, 2023. Data were accessed for research purposes, and data collection began on February 1, 2024.

The participating hospitals comprise three private, university-affiliated hospitals and one public teaching hospital. The private hospitals provide medical services to individuals from diverse socioeconomic backgrounds, while the public hospital primarily serves low-to-middle-income patients. These institutions offer specialized healthcare services, collectively addressing a broad spectrum of medical needs within the Lebanese community. The bed capacities vary, with approximately 75 beds in one hospital, 97 beds in another, and 250 beds each in the third and fourth hospitals.

All participating hospitals have Infection Prevention and Control (IPC) committees and, to varying extents, AMS programs. However, the structure, implementation, AMR containment efforts, and clinician compliance differ among the participating institutions. The private hospitals have well-established AMS and IPC committees, consisting of infectious disease specialists, clinical pharmacists, microbiologists, infection control officers, laboratory technicians, information technology specialists, and hospital epidemiologists. These committees oversee antibiotic prescribing practices, infection control measures, compliance with standard and transmission-based precautions, and proper sterilization procedures for hospital units and medical equipment. Additionally, they facilitate patient and visitor education, conduct epidemiological studies, and provide training sessions for healthcare professionals.

In contrast, the public hospital has an IPC committee composed of infectious disease specialists and nursing staff but lacks a formal AMS committee. This results in the absence of structured AMS protocols, antimicrobial susceptibility surveillance reports, and the enforcement of antimicrobial prescription through preauthorization policies.

## 2.2. *Study population*

The study included hospitalized adult patients aged 18 years and older with a confirmed diagnosis of *P. aeruginosa* infection who received treatment at any of the participating hospitals during the study period. For patients with recurrent *P. aeruginosa* infections, only the first infection episode was documented to avoid overrepresentation of individual patients in the dataset and to reflect baseline antimicrobial resistance profiles without the confounding influence of prior treatments or recurrent infections.

The exclusion criteria were as follows: cases of colonization rather than active infection, polymicrobial infections or co-infections, patients treated as outpatients (i.e., those not admitted to the hospital), subsequent hospitalizations due to recurrent infections, patients discharged or who left the hospital before receiving initial antibiotic therapy, and incomplete medical records lacking antimicrobial susceptibility testing (AST) results or other crucial medical information. These criteria were designed to establish a focused and representative study population of hospitalized patients with active *P. aeruginosa* infections. By excluding cases involving recurrent infections, co-infections, or colonization, the study aimed to minimize confounding factors that could obscure the specific impact of *P. aeruginosa* on clinical outcomes. This approach enabled a targeted evaluation of the appropriateness of initial antibiotic therapy for infections caused by *P. aeruginosa* and facilitated the estimation of the associated in-hospital mortality.

## 2.3. *Microbiological identification and antimicrobial susceptibility testing*

Microbiological identification and AST of *P. aeruginosa* were conducted at four participating hospitals, each following its respective laboratory protocols. In Hospitals 1 and 2, bacterial identification and AST were performed using the VITEK 2 automated system (bioMérieux, Marcy-l'Étoile, France), which determines the minimum inhibitory concentrations (MICs) of tested antibiotics. In contrast, Hospitals 3 and 4 employed the Kirby-Bauer disk diffusion method, following the Clinical and Laboratory Standards Institute (CLSI) guidelines. Inhibition zone diameters were measured manually and verified using the ADAGIO automated zone reader (Bio-Rad Laboratories, Hercules, CA, USA).

AST results for all hospitals were interpreted according to CLSI-defined breakpoint criteria [19]. The susceptibility of *P. aeruginosa* isolates was classified as susceptible (S), intermediate (I), or resistant (R).

Routine AST included a panel of β-lactams (piperacillin-tazobactam, ceftazidime, cefepime, meropenem or imipenem, and aztreonam), aminoglycosides (gentamicin, tobramycin, and amikacin), fluoroquinolones (ciprofloxacin), and polymyxins (colistin). In cases where resistance to standard β-lactams was detected, additional AST was performed in one hospital against novel β-lactam/β-lactamase inhibitor combinations, including ceftazidime-avibactam and ceftolozane-tazobactam, to assess their potential efficacy against resistant strains.

## 2.4. *Data collection and key definitions*

The microbiology laboratories of the participating hospitals provided the study's principal investigator with case numbers of patients whose clinical specimens grew *P. aeruginosa* isolates during the study period. Medical records corresponding to these case numbers were meticulously reviewed to determine eligibility based on the study's inclusion criteria. For eligible patients, relevant medical data were collected, including demographics and clinical characteristics such as age, sex, total hospital stay duration, and infection onset. Infection onset was defined as the day the clinical specimen grew *P. aeruginosa* [20]. Infections were categorized as either community-acquired (manifesting before hospitalization or within 48 hours of admission) or hospital-acquired (manifesting 48 hours or more after admission, or within seven days post-discharge) [20]. Patient comorbidities were documented, and the Charlson Comorbidity Index (CCI) score was calculated for each patient [21,22]. Additionally, prior antibiotic use or hospitalization within the three months preceding the infection was recorded, along with any glucocorticoid administration in the past four weeks or chemotherapy in the preceding three months.

The site of infection acquisition as well as the microbiological data, including the AST results were documented. *P. aeruginosa* isolates were categorized based on their resistance profiles into three groups: multidrug-resistant (MDR), defined as "non-susceptibility to at least one agent in three or more antimicrobial categories"; extensively drug-resistant (XDR), defined as "non-susceptibility to at least one agent in all but two or fewer antimicrobial categories"; and pandrug-resistant (PDR), defined as "non-susceptibility to all agents in every antimicrobial category" [23]. Additionally, isolates were classified as DTR *P. aeruginosa* and carbapenem-resistant *P. aeruginosa* (CRPA). DTR *P. aeruginosa* is defined as "isolates non-susceptible (either resistant or intermediate) to all of the following agents: piperacillin-tazobactam, ceftazidime, cefepime, aztreonam, meropenem, imipenem-cilastatin, ciprofloxacin, and levofloxacin" [8]. Conversely, isolates that do not meet these criteria are classified as non-DTR *P. aeruginosa*. CRPA is defined as "isolates resistant to imipenem-cilastatin and/or meropenem" [8].

Details regarding the antibiotic therapies administered to treat patients' infections during their hospital stay were documented, including both initial and definitive therapies. The following information was recorded for each antibiotic treatment: the specific antibiotics used, the date of initiation relative to infection onset, the route of administration, the dosage regimen, and the duration of treatment. Initial antibiotic therapy, or empirical therapy, refers to the antibiotics administered before the causative pathogen is identified and before AST results are available. In contrast, definitive therapy involves the antibiotics administered after the causative pathogen has been identified and AST results obtained [20].

The appropriateness of the initial antibiotic therapy was assessed by determining whether one or more antibiotic agents were administered within 48 hours of infection onset, and whether the causative pathogen was susceptible to these agents according to AST results. The antibiotics must have been delivered via the correct route and at an appropriate dosage regimen. Any initial antibiotic therapy that did not meet these criteria was considered inappropriate [20].

The infection prognosis, associated complications, and treatment outcomes were documented. The primary outcome of interest was 30-day all-cause in-hospital mortality. Additionally, the time-to-mortality, defined as the interval between infection onset and death, was recorded [20].

## 2.5. *Ethical considerations*

The study was conducted in accordance with the ethical principles of the World Medical Association's Declaration of Helsinki [24]. The research protocol was approved by the Institutional Review Boards (IRB) of the participating hospitals: first hospital (approval code: 2023-IRB-012), second hospital (approval code: 8785/AM/T/BDM/Classification 30350/1), third hospital (approval code: 6/2023), and fourth hospital (approval code: HOP-2023–006). Data collection involved a comprehensive review of medical records containing identifiable information on individual participants. However, the research team did not contact or follow up with patients. Given the retrospective and observational nature of the study, which utilized post-discharge data while ensuring participant confidentiality, the IRBs of the participating hospitals granted a waiver of informed consent.

## 2.6. *Statistical analysis*

Data were analyzed using version 24 of the Statistical Package for the Social Sciences (SPSS®, IBM Corp., Armonk, NY, USA). Descriptive statistics were reported as frequencies and percentages for categorical variables, and as means with standard deviations for continuous variables. Kaplan-Meier survival analysis was used to examine time-to-mortality, with the log-rank ($\chi^2$) test applied to compare survival times between two groups: patients infected with DTR *P. aeruginosa* isolates and those infected with non-DTR isolates. This test also evaluated differences in survival times between patients who received appropriate versus inappropriate initial antibiotic therapy.

Univariate analysis was conducted to assess associations between independent variables and 30-day all-cause mortality, utilizing Pearson's chi-square ($\chi^2$) test for categorical variables and independent t-tests for continuous variables. Variables with a P-value $< 0.2$ in the univariate analysis were included in the multivariable logistic regression model. The

variables initially included were: age group (≥ 65 years vs. 18–64 years), sex (female vs. male), site of infection acquisition (community-acquired vs. nosocomial infection), hospital department of infection acquisition (ICU vs. non-ICU), coronary artery disease, congestive heart failure, CCI, previous glucocorticoid use within the last four weeks, type of antimicrobial resistance (DTR vs. non-DTR *P. aeruginosa* infection), and appropriateness of the initial antibiotic therapy (appropriate vs. inappropriate).

Multicollinearity was assessed using variance inflation factors (VIF), with a threshold of 10 for exclusion. All VIF values were below 2.5, indicating no significant multicollinearity among the predictors. Conceptually related variables were carefully reviewed for inclusion based on clinical relevance and contribution to model fit. After careful consideration, ICU vs. non-ICU was retained for its stronger relevance to patient outcomes, while nosocomial vs. community-acquired infection was excluded. Similarly, congestive heart failure and coronary artery disease were excluded due to their conceptual overlap with CCI, which was retained as a more comprehensive measure of comorbidity burden.

The final multivariable logistic regression model was developed using a stepwise forward selection approach, which sequentially added variables based on their statistical contribution to the model. Confounding variables were retained in the model if their inclusion altered the coefficient of any significant variable by 10% or more. The glucocorticoid use, although identified in the univariate analysis ($P < 0.2$), was excluded during forward selection as it did not significantly contribute to the final model after accounting for other variables. The goodness-of-fit for the model was assessed using Hosmer-Lemeshow test. Adjusted odds ratios (AOR) with 95% confidence intervals and P-values were calculated for all variables in the model.

## 3. Results

### 3.1. *Case selection and enrollment*

A total of 2,639 cases with confirmed *P. aeruginosa* isolates were initially reviewed. Of these, 2,162 cases were excluded for not meeting the inclusion criteria. Reasons for exclusion included: 626 patients had polymicrobial infections or co-infections during the same admission; 389 patients were treated as outpatients; 346 cases involved colonization rather than active infection; 302 patients were under 18 years of age, 283 cases involved recurrent infections (only the first episode was considered); 129 patients left the hospital within 48 hours of infection acquisition before receiving initial antibiotic therapy; and 87 patients had incomplete medical records (Fig 1).

As a result, 477 patients with *P. aeruginosa* infections were included in this study. These patients were analyzed for antimicrobial susceptibility patterns and the appropriateness of the initial antibiotic therapy. However, the final analysis of treatment outcomes and mortality predictors included only 465 patients, as twelve patients left the hospital against medical advice and were subsequently lost to follow-up.

### 3.2. *Patient baseline characteristics and clinical profiles*

Nearly two-thirds of the patients (296, 62.1%) were elderly, aged 65 years or older, and the majority were male (294, 61.6%). Almost half of the infections (227, 47.6%) were hospital-acquired, with more than half of these (118/227, 52%) originating in intensive care units (ICU). The mean duration of hospitalization was 16.35±12.5 days, ranging from 3 to 76 days.

Most patients (431, 90.4%) had at least one underlying medical condition, with hypertension being the most common (246, 51.6%), followed by type II diabetes mellitus (198, 41.5%), coronary artery disease (146, 30.6%), and dyslipidemia (108, 22.6%). The mean CCI was 4.46±2.43, with scores ranging from 0 to 11.

Approximately one-quarter of the patients (116, 24.3%) had received antibiotics prior to acquiring the infection, while 22.2% had been hospitalized within the three months preceding infection. Additionally, less than one-fifth of the patients (81, 17%) had used glucocorticoids in the past four weeks (95, 19.9%) or had undergone chemotherapy in the previous three months.

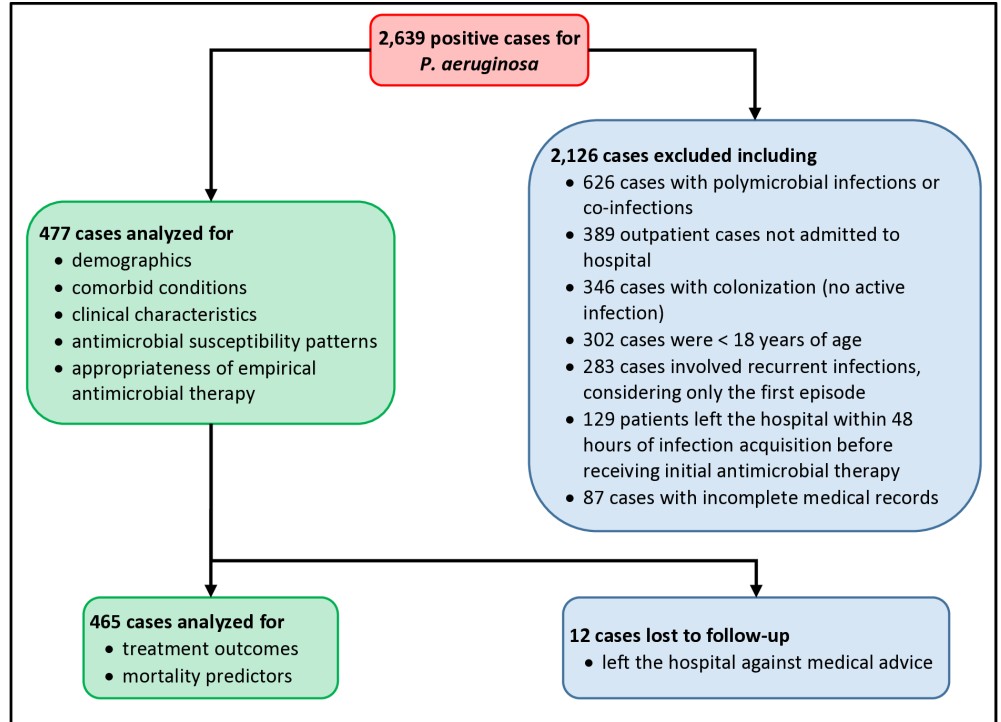

**Fig 1. Flowchart depicting the enrollment process of patients diagnosed with *Pseudomonas aeruginosa* infection and the subsequent data analysis.**

### 3.3. *Infection site distribution and resistance profiles of Pseudomonas aeruginosa isolates*

The respiratory tract was the most common primary site of infection, accounting for 185 cases (38.8%), followed by skin and soft tissue with 156 cases (32.7%). Other infection sites included the urinary tract (98, 20.5%), bloodstream (18, 3.8%), intra-abdominal (10, 2.1%), surgical sites (8, 1.7%), and bone (2, 0.4%).

Approximately 27.5% of the isolates were resistant to meropenem, with slightly lower resistance rates observed for imipenem (24.4%), aztreonam (22.7%), and ciprofloxacin (20.4%). Resistance to piperacillin-tazobactam, ceftazidime, and cefepime was observed in 16.3%, 15.6%, and 11.8% of the isolates, respectively. Notably, two isolates (0.4%) showed resistance to colistin. In contrast, the majority of isolates remained susceptible to tobramycin (91.6%), amikacin (89.9%), and gentamicin (87.2%). Out of the 28 isolates tested for ceftazidime-avibactam, 10 isolates (10/28, 35.7%) were resistant. Meanwhile, all six isolates tested for ceftolozane-tazobactam (6/6, 100%) were susceptible (S1 Table).

Among the isolates, 29.8% were classified as MDR, and 16.6% were XDR. None of the isolates were PDR. Additionally, 29.9% of the isolates were identified as CRPA, while 15.3% were categorized as DTR.

Among the CRPA isolates, 50.4% remained susceptible to ceftazidime, 48.2% to cefepime, and 40.3% to piperacillin/tazobactam.

A significant association was observed between specific infection sites and the acquisition of DTR *P. aeruginosa* strains. Specifically, infections originating from bloodstream (odds ratio [OR] = 2.93, 95% CI = 1.06–8.06, P = 0.03) and respiratory tract (OR = 2.64, 95% CI = 1.59–4.40, P < 0.01) were independently associated with an increased likelihood of acquiring a DTR infection.

### 3.4. Assessment of received antibiotic therapy and its association with difficult-to-treat resistant *Pseudomonas aeruginosa* infections

Of the 477 enrolled patients, one-third (156, 32.7%) received combination therapy. Nearly all initial therapies were administered intravenously (IV) (465, 97.5%), with a mean time of 1.28±0.86 days from infection onset to the initiation of antibiotic therapy. The most common empirical antibiotics were piperacillin-tazobactam (162, 33.9%) and meropenem (117, 24.5%). A single dose of amikacin was used as adjunctive treatment in 5.8% of the cases. Detailed information on empirical antibiotic therapy is provided in **Table 1**.

Nearly half of the patients (209, 43.8%) received inappropriate initial antibiotic therapy. Specifically, one-third of these patients (161, 33.8%) were administered antibiotics to which the causative pathogen was not sensitive, based on AST results. Additionally, 30 patients (6.3%) experienced a delay in receiving antibiotic therapy beyond 48 hours from the onset of infection, and 34 patients (7.1%) were given inappropriate doses, either subtherapeutic or with incorrect dosing frequencies. However, all patients received their therapy through the correct route of administration.

Patients infected with DTR *P. aeruginosa* were significantly more likely to receive inappropriate initial antibiotic therapy compared to those infected with non-DTR isolates (OR=4.21, 95% CI=2.43–7.32, P<0.001).

Among the total cohort (N=477), two patients (0.4%) died while receiving empirical antibiotic therapy, and 12 patients (2.5%) left the hospital against medical advice, resulting in a loss to follow-up. Of the remaining hospitalized patients (N=463), the majority (311, 67.1%) were either deescalated or switched to definitive antibiotic therapy after culture and AST results were obtained. The rest (152, 32.8%) continued on empirical therapy throughout their hospital stay.

For patients whose therapy was deescalated or shifted to definitive therapy (N=311), the mean time from obtaining AST results to implementing definitive treatment was 1.85±1.49 days. Notably, 18.6% (58/311) experienced delays of more than 24 hours in adjusting their therapy after receiving AST results.

It is worth noting that 16 patients (3.5%) were given antibiotics to which the causative pathogen was not sensitive, even after culture results were available. The most frequently used antibiotics after obtaining AST results were ciprofloxacin (118, 25.5%), piperacillin-tazobactam (112, 24.2%), and meropenem (76, 16.4%). Interestingly, nearly half of the patients (220, 47.5%) continued receiving antibiotics that did not align with the narrowest spectrum agents available, despite the availability of narrower alternatives to which the pathogen was sensitive. Additionally, 6.9% of patients were prescribed antibiotics deemed unsafe due to drug and patient-specific factors.

Only a small proportion of patients (94, 20.3%) were transitioned to oral antibiotics, while the majority (365, 78.8%) remained on IV therapy throughout their hospital stay. For more details on definitive antibiotic therapy, refer to **Table 2**.

### 3.5. Infection prognosis, associated complications, and treatment outcomes

After excluding the 12 patients lost to follow-up, the remaining 465 patients were analyzed for infection prognosis and treatment outcomes. Nearly 30% of the patients (136, 29.2%) developed sepsis or septic shock following the acquisition of *P. aeruginosa* infection, with a mean sequential organ failure assessment (SOFA) score of 5.89±2.54, ranging from 3 to 13.

Several patients also experienced additional complications, including acute kidney injury (32, 6.8%), pleural effusion (30, 6.4%), atelectasis (21, 4.5%), electrolyte disturbances (16, 3.4%), neutropenia (10, 2.1%), bronchiectasis (9, 1.9%), acid-base imbalances (9, 1.9%), thrombocytopenia (8, 1.7%), disseminated intravascular coagulation (7, 1.5%), cardiomegaly (7, 1.5%), acute respiratory distress syndrome (6, 1.2%), pneumothorax (5, 1%), and renal abscess formation (1, 0.2%).

Unfortunately, 69 patients (14.8%) died during their hospital stay within 30 days of infection acquisition, including the two patients who died while receiving empirical antibiotic therapy.

**Table 1. Empirical antibiotic therapy received (N = 477).**

| Information | n (%) |
|---|---|
| **Onset of therapy initiation with respect to the onset of infection** | |
| *Mean = 1.28 ± 0.86 days (range = 1–7)* | |
| **Route of administration** | |
| *Intravenous* | 465 (97.5) |
| *Oral* | 6 (1.3) |
| *Inhaled* | 4 (0.8) |
| *Intramuscular* | 2 (0.4) |
| **Monotherapy vs. combination therapy** | |
| *Monotherapy* | 321 (67.3) |
| *Combination* | 156 (32.7) |
| **Antibiotic agents used**[a] | |
| *Piperacillin-tazobactam* | 162 (33.9) |
| *Meropenem* | 117 (24.5) |
| *Ceftriaxone* | 38 (7.9) |
| *Ciprofloxacin* | 38 (7.9) |
| *Colistin* | 30 (6.2) |
| *Amikacin* | 28 (5.8) |
| *Clindamycin* | 28 (5.8) |
| *Vancomycin* | 25 (5.2) |
| *Imipenem-cilastatin* | 23 (4.8) |
| *Cefepime* | 22 (4.6) |
| *Teicoplanin* | 18 (3.7) |
| *Levofloxacin* | 17 (3.5) |
| *Ceftazidime* | 16 (3.3) |
| *Moxifloxacin* | 12 (2.5) |
| *Linezolid* | 12 (2.5) |
| *Clarithromycin* | 10 (2) |
| *Cefazolin* | 8 (1.6) |
| *Ceftaroline* | 7 (1.4) |
| *Metronidazole* | 6 (1.2) |
| *Azithromycin* | 6 (1.2) |
| *Ceftazidime-avibactam* | 6 (1.2) |
| *Amoxicillin-clavulanic acid* | 5 (1.0) |
| *Ertapenem* | 4 (0.8) |
| *Trimethoprim-sulfamethoxazole* | 4 (0.8) |
| *Cefixime* | 2 (0.4) |
| *Tigecycline* | 2 (0.4) |
| *Cefuroxime* | 1 (0.2) |
| **Assessment of appropriateness** | |
| ***Sensitive to the culture results*** | |
| No | 161 (33.8) |
| Yes | 316 (66.2) |
| ***Initiated within 48 hours of the onset of infection*** | |
| No | 30 (6.3) |
| Yes | 447 (96.7) |

*(Continued)*

**Table 1.** (Continued)

| Information | n (%) |
|---|---|
| *Given for at least 48 hours* | |
| No | 5 (1) |
| Yes | 472 (99) |
| *Given with an appropriate dose* | |
| No | 34 (7.1) |
| Yes | 443 (92.9) |
| *Given through an appropriate route of administration* | |
| No | 0 (0) |
| Yes | 477 (100) |
| *Overall appropriateness* | |
| No | 209 (43.8) |
| Yes | 268 (56.2) |

[a]Multiple individual responses were recorded, so the total count does not equal 477.

### 3.6. *Mortality associated with Pseudomonas aeruginosa infections*

Twelve patients left the hospital against medical advice and were lost to follow-up, with their treatment outcomes not documented in medical records. Among the remaining 465 patients, the 30-day all-cause mortality rate due to *P. aeruginosa* infection was 14.8% (69 patients). The mean time-to-mortality was $13.29 \pm 9.81$ days (**Fig 2**).

### 3.7. *Comparison of survival of patients with Pseudomonas aeruginosa infections based on the appropriateness of initial antibiotic therapy*

**Fig 3** presents the survival analysis curve for *P. aeruginosa* infections based on the appropriateness of the initial antibiotic therapy received. The analysis showed a significantly shorter time to mortality in patients who received inappropriate initial antibiotic therapy compared to those who received appropriate therapy, with 41 and 28 deaths out of 69, respectively (log-rank test $= 4.48$, $P = 0.03$). The mean time to death for patients receiving inappropriate antibiotic therapy was $11.76 \pm 8.80$ days, compared to $15.46 \pm 10.90$ days for those receiving appropriate therapy. (**Table 3**).

### 3.8. *Comparison of survival between patients who died from infection with difficult-to-treat resistant vs. non-difficult-to-treat resistant Pseudomonas aeruginosa isolates*

The survival analysis showed a non-significant trend toward shorter time to mortality in patients infected with DTR *P. aeruginosa* compared to those with non-DTR isolates (log-rank test $= 3.08$, $P = 0.07$) (**Fig 4**). The mean time to death in the DTR group was $10.41 \pm 9.87$ days, compared to $14.34 \pm 9.68$ days in the non-DTR group (Table 4).

### 3.9. *Mortality predictors in patients with Pseudomonas aeruginosa infections*

**Table 5** presents the univariate analysis of potential predictors associated with mortality in patients infected with *P. aeruginosa*. The analysis revealed that mortality was significantly associated with *P. aeruginosa* isolates acquired from the ICU (UOR $= 4.82$, 95% CI $= 2.84$–$8.26$, $P < 0.001$) and infection with DTR *P. aeruginosa* isolates (UOR $= 3.46$, 95% CI $= 1.93$–$6.19$, $P < 0.01$). Similarly, nosocomial infection was linked to an increased risk of mortality (UOR $= 3.06$, 95% CI $= 1.76$–$5.34$, $P < 0.001$). Underlying health conditions, including congestive heart failure (UOR $= 2.11$, 95% CI $= 1.03$–$4.28$, $P = 0.03$) and a higher CCI ($P = 0.01$), were also significantly associated with mortality. Additionally, female sex (UOR $= 2.07$, 95% CI $= 1.23$–$3.46$, $P = 0.004$) and glucocorticoid use within the last four weeks (UOR $= 1.90$, 95% CI $= 1.07$–$3.37$, $P = 0.02$)

**Table 2. Definitive antibiotic therapy administered (N = 463)[a].**

| Information | n (%) |
|---|---|
| **Therapy was shifted or de-escalated to definitive antibiotic therapy** | |
| *No, kept on empirical therapy* | 152 (32.8) |
| *Yes* | 311 (67.1) |
| **Onset of therapy initiation relative to obtaining the culture results** | |
| *Mean = 1.85 ± 1.49 days (range = 1–9)* | |
| **Route of administration** | |
| *Intravenous* | 365 (78.8) |
| *Oral* | 94 (20.3) |
| *Inhaled* | 4 (0.9) |
| **Monotherapy vs. combination therapy** | |
| *Monotherapy* | 419 (90.5) |
| *Combination* | 44 (9.5) |
| **Antibiotic agents used[b]** | |
| *Ciprofloxacin* | 118 (25.5) |
| *Piperacillin-tazobactam* | 112 (24.2) |
| *Meropenem* | 76 (16.4) |
| *Colistin* | 51 (11) |
| *Cefepime* | 36 (7.8) |
| *Levofloxacin* | 28 (6) |
| *Ceftazidime-avibactam* | 18 (3.9) |
| *Amikacin* | 14 (3) |
| *Ceftazidime* | 14 (3) |
| *Imipenem-cilastatin* | 10 (2.2) |
| *Fosfomycin* | 4 (0.9) |
| **Assessment of appropriateness** | |
| *Sensitive to the culture results* | |
| No | 16 (3.5) |
| Yes | 447 (96.5) |
| *Given with an appropriate dose* | |
| No | 14 (3) |
| Yes | 449 (97) |
| *Appropriate route of administration* | |
| No | 0 (0) |
| Yes | 100 (100) |
| *According to the antibiogram results, the selected antibiotic was the narrowest among the ones the bacteria was sensitive to* | |
| No, there are narrower alternatives that were also sensitive to the isolated pathogen | 220 (47.5) |
| Yes | 243 (52.5) |
| *The selected antibiotic was safe according to drug and patient-specific factors* | |
| No, there were safer alternatives | 32 (6.9) |
| No, but there were no safer alternatives | 20 (4.3) |
| Yes | 409 (88.7) |

[a]Fourteen out of the initial 477 patients either died or were lost to follow-up while receiving empirical therapy.

[b]The total of 463 is not reflected in the numbers due to some patients having multiple responses recorded.

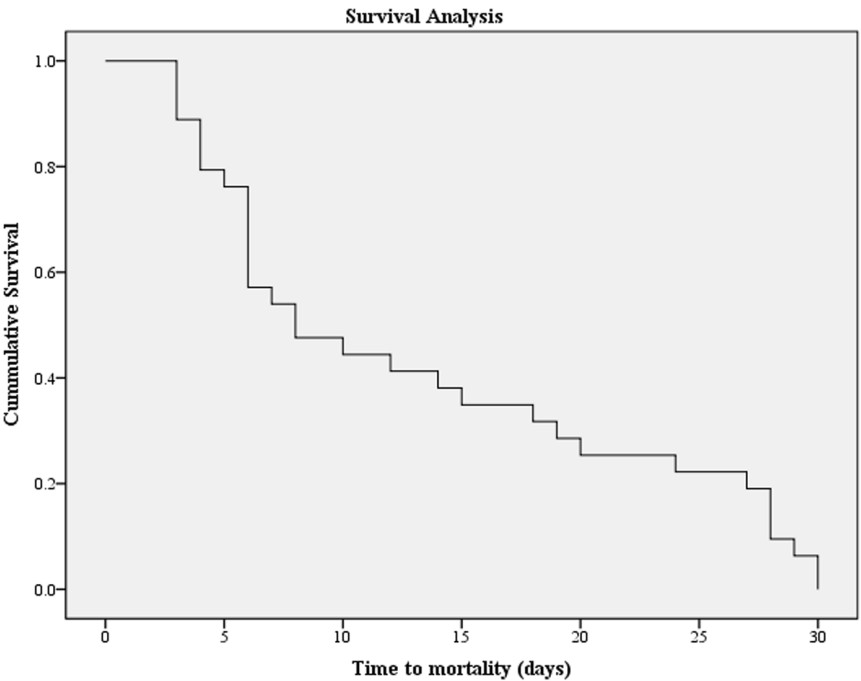

**Fig 2. The Kaplan-Meier survival curve for patients infected with *Pseudomonas aeruginosa*.**

were identified as risk factors. Age ≥ 65 years was also associated with an increased risk of mortality (UOR = 1.75, 95% CI = 0.99–3.07, P = 0.04). Finally, receiving inappropriate initial antibiotic therapy significantly contributed to a higher mortality risk (UOR = 2.07, 95% CI = 1.23–3.49, P = 0.005).

Binary logistic regression identified two main predictors significantly associated with mortality in patients with *P. aeruginosa* infection (**Table 6**). These predictors were infection with DTR *P. aeruginosa* isolates (AOR = 2.48, 95% CI = 1.32–4.63, P < 0.01), and receiving inappropriate initial antibiotic therapy (AOR = 1.40, 95% CI = 1.04–2.35, P < 0.01).

## 4. Discussion

This study revealed a high level of resistance in *P. aeruginosa* isolates to carbapenems, with approximately 30% of the isolates identified as CRPA. This finding aligns with national studies conducted between 2010 and 2019, which reported a similar mean CRPA prevalence of 30% ± 3% across tertiary care hospitals in Lebanon [2,18,25–29]. Despite the consistently high CRPA rate in Lebanon, there has been no clear trend indicating either an increase or decrease over the years.

On a regional level, the prevalence of CRPA varies significantly across the Middle East. For example, Jordan reported a prevalence of 41.9% between 2020 and 2022 [30], while Iraq exhibited a notably high rate of 68.4% during the same period [31]. In comparison, Saudi Arabia reported CRPA rates ranging between 29.5% and 37.2% during 2021 and 2022 [32,33]. Meanwhile, the United Arab Emirates demonstrated a much lower CRPA prevalence, ranging from 13% to 14.1% between 2010 and 2021 [34]. Globally, a recent study analyzing AMR trends in 30,504 *P. aeruginosa* clinical isolates reported CRPA rates ranging from 19% to 30% between 2018 and 2022. Notably, the study highlighted a significant decrease in CRPA prevalence in the Middle East and Africa, dropping from 28% in 2018 to 22.4% in 2022 [9]. Lebanon's CRPA rate, while aligned with the higher end of the global spectrum, underscores the need for stronger infection control measures and regulatory oversight, especially in light of the persistent overuse of carbapenems. Neighboring countries that have successfully reduced resistance rates could provide valuable insights into strategies Lebanon might adopt to curb this issue.

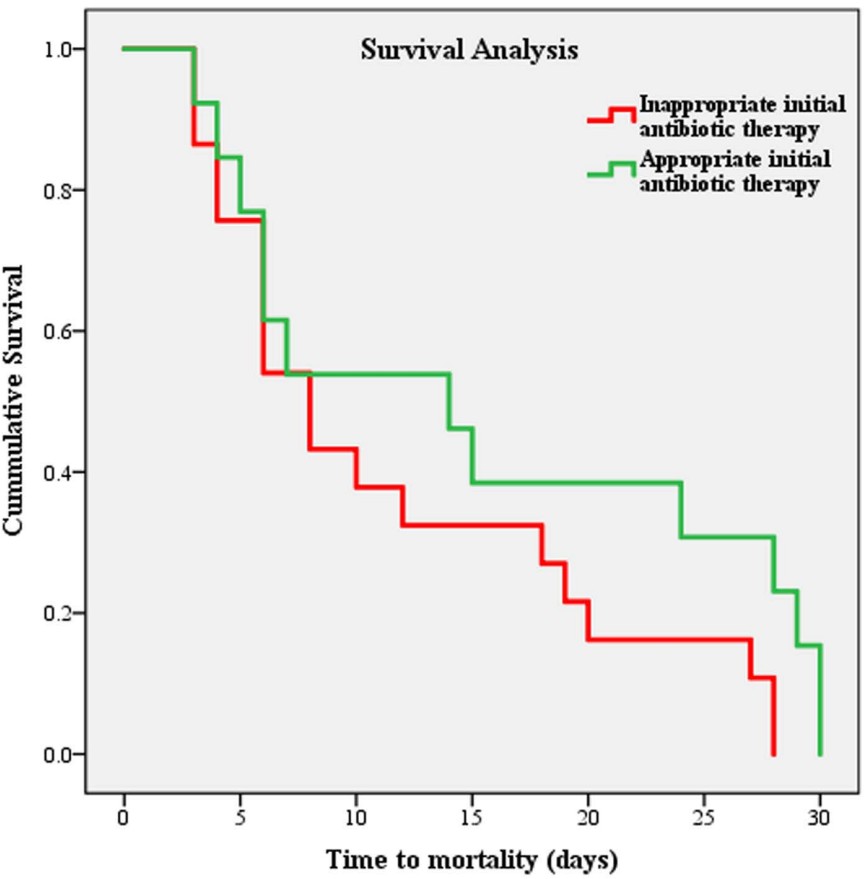

**Fig 3. The Kaplan-Meier survival curves for patients with *Pseudomonas aeruginosa* infection based on the appropriateness of the initial antibiotic therapy received.**

**Table 3. Comparison of survival time between patients who died after receiving appropriate antibiotic therapy and those who died after receiving inappropriate antibiotic therapy (N = 69 out of 465).**

| Group | 30-day mortality n (%) | Mean survival time (days) Mean ± SD | Mean 95% CI | Log-rank $X2$ test | P |
|---|---|---|---|---|---|
| **Appropriate antibiotic therapy** (n = 260) | 28 (40.5) | 15.46 ± 10.90 | 11.27–19.65 | 4.48 | 0.03[*] |
| **Inappropriate antibiotic therapy** (n = 205) | 41 (59.5) | 11.76 ± 8.80 | 8.92–14.59 | | |

*CI*, confidence interval; *SD*, standard deviation.

* Statistically significant (< 0.05).

Nearly half of the CRPA isolates in this study exhibited concomitant susceptibility to broad-spectrum cephalosporins (e.g., cefepime, ceftazidime) and piperacillin/tazobactam. Consistent with our findings, a recent national multicenter epidemiological study conducted among 2,400 patients diagnosed with Gram-native bacteremia reported higher resistance rates of *P. aeruginosa* to carbapenems while showing lower resistance to ceftazidime and cefepime [35]. This resistance phenotype is primarily attributed to the mutational inactivation or downregulation of the outer membrane porin (OprD), which selectively facilitates the uptake of carbapenems into the bacterial cell. Mutations or reduced expression of OprD decrease carbapenem influx, conferring resistance while preserving the efficacy of non-carbapenem antipseudomonal β-lactams, such as cefepime and piperacillin/tazobactam, as their uptake pathways remain unaffected [36,37]. Similarly,

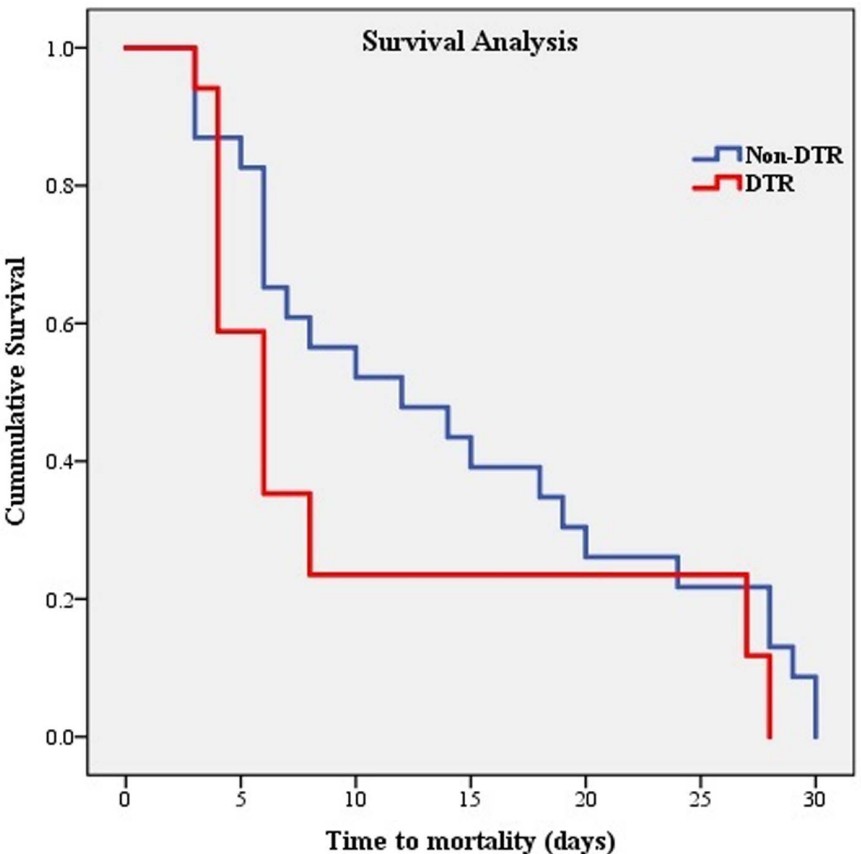

**Fig 4. The Kaplan-Meier survival curves for patients with difficult-to-treat resistant (DTR) and non-DTR *Pseudomonas aeruginosa* infection.**

**Table 4. Comparison of survival times between patients infected with difficult-to-treat resistant and non-difficult-to-treat *Pseudomonas aeruginosa* isolates (N = 69 out of 465).**

| Group | 30-days mortality n (%) | Mean survival time (days) Mean ± SD | Mean 95% CI | Log-rank $X2$ test | P |
|---|---|---|---|---|---|
| **DTR** (n = 73) | 23 (33.3) | 10.41 ± 9.87 | 5.71–15.10 | 3.08 | 0.07* |
| **Non-DTR** (n = 392) | 46 (66.7) | 14.34 ± 9.68 | 11.55–17.14 | | |

*CI*, confidence interval; *DTR*, difficult-to-treat resistance; *SD*, standard deviation.
* Statistically not significant ($p > 0.05$).

a national study investigating the molecular mechanisms of CRPA revealed that OprD gene disruption, caused by either mutations or insertion sequence elements, was a major determinant of imipenem resistance [38]. Additionally, the over-expression of efflux pumps, particularly MexAB-OprM or MexXY-OprM, contributes to this resistance profile. These efflux systems selectively expel carbapenems more efficiently than other β-lactams, leading to carbapenems resistance while maintaining susceptibility to other antipseudomonal agents [39]. Further molecular epidemiological studies are needed to confirm this observed resistance pattern across different hospitals in Lebanon. These findings are crucial for optimizing

**Table 5. Univariate analysis of predictors associated with mortality in patients with *Pseudomonas aeruginosa* infection (N = 465)[a].**

| Characteristics | n (%)[b] | 30-Day all-cause mortality | | UOR (95% CI) | Pearson's χ² | P[d] |
|---|---|---|---|---|---|---|
| | | Survived (396) n (%)[c] | Died (69) n (%)[c] | | | |
| **Age group[†] (reference: 18–64 years)** | | | | | | |
| 18–64 | 177 (38.1) | 158 (89.3) | 19 (10.7) | – | 3.80 | 0.04[e] |
| ≥ 65 | 288 (61.9) | 238 (82.6) | 50 (17.4) | 1.75 (0.99–3.07) | | |
| **Sex[†] (reference: female)** | | | | | | |
| Male | 286 (61.5) | 254 (88.8) | 32 (11.2) | – | 7.83 | 0.004[e] |
| Female | 179 (38.5) | 142 (79.3) | 37 (20.7) | 2.07 (1.23–3.46) | | |
| **Site of infection acquisition (reference: community-acquired)** | | | | | | |
| Community-acquired | 240 (51.6) | 220 (91.7) | 20 (8.3) | – | 16.6 | <0001[e] |
| Nosocomial | 225 (48.4) | 176 (78.2) | 49 (21.8) | 3.06 (1.76–5.34) | | |
| **Hospital department of infection acquisition[†] (reference: non-ICU)** | | | | | | |
| Non-ICU | 347 (74.6) | 316 (91.1) | 31 (8.9) | – | 37.73 | <0.001[e] |
| ICU | 118 (25.4) | 80 (67.8) | 38 (32.2) | 4.84 (2.84–8.26) | | |
| **Underlying medical conditions (reference: no history of the related condition)** | | | | | | |
| Hypertension | 238 (51.2) | 198 (83.2) | 40 (16.8) | 1.38 (0.82–2.31) | 1.49 | 0.22 |
| Type II diabetes mellitus | 190 (40.9) | 164 (86.3) | 26 (13.7) | 0.86 (0.51–1.45) | 0.33 | 0.56 |
| Coronary artery disease | 144 (31) | 128 (88.9) | 16 (11.1) | 0.63 (0.35–1.15) | 2.29 | 0.13 |
| Dyslipidemia | 104 (22.4) | 88 (84.6) | 16 (15.4) | 1.06 (0.58–1.94) | 0.03 | 0.85 |
| Chronic obstructive pulmonary disease | 72 (15.5) | 62 (86.1) | 10 (13.9) | 0.91 (0.44–1.88) | 0.06 | 0.80 |
| Chronic kidney disease | 58 (12.5) | 52 (89.7) | 6 (10.3) | 0.63 (0.26–1.53) | 1.05 | 0.30 |
| Cancer/Solid tumor (localized) | 57 (12.3) | 48 (84.2) | 9 (15.8) | 1.09 (0.51–2.33) | 0.04 | 0.82 |
| Congestive heart failure | 48 (10.3) | 36 (75) | 12 (25) | 2.11 (1.03–4.28) | 4.37 | 0.03[e] |
| **Charlson Comorbidity Index[†]** | | | | | | |
| Mean ± SD | 4.48 ± 2.44 | 4.38 ± 2.41 | 6.04 ± 2.55 | – | – | 0.01[e,f] |
| **Previous hospitalization within the last three months (reference: no)** | | | | | | |
| No | 359 (77.2) | 306 (85.2) | 53 (14.8) | – | 0.007 | 0.93 |
| Yes | 106 (22.8) | 90 (84.9) | 16 (15.1) | 1.03 (0.56–1.88) | | |
| **Previous antibiotics use within the last three months (reference: no)** | | | | | | |
| No | 351 (75.5) | 302 (86) | 49 (14) | – | 0.87 | 0.35 |
| Yes | 114 (24.5) | 94 (82.5) | 20 (17.5) | 1.31 (0.74–2.32) | | |
| **Previous glucocorticoids use within the last four weeks (reference: no)** | | | | | | |
| No | 370 (79.6) | 322 (87) | 48 (13) | – | 4.98 | 0.02[e] |
| Yes | 95 (20.4) | 74 (77.9) | 21 (22.1) | 1.90 (1.07–3.37) | | |
| **Previous chemotherapy use within the last three months (reference: no)** | | | | | | |
| No | 384 (82.6) | 328 (85.4) | 56 (14.6) | – | 0.11 | 0.73 |
| Yes | 81 (17.4) | 68 (84) | 13 (16) | 1.12 (0.58–2.16) | | |
| **Source of infection (reference: urinary tract)** | | | | | | |
| Respiratory tract | 183 (39.4) | 150 (82) | 33 (18) | 1.89 (0.89–4.03) | 6.12 | 0.29 |
| Soft tissue/wound | 152 (32.7) | 134 (88.2) | 18 (11.8) | 1.16 (0.51–2.62) | | |
| Urinary tract | 96 (20.6) | 86 (89.6) | 10 (10.4) | – | | |
| Bloodstream | 18 (3.9) | 14 (77.8) | 4 (22.2) | 2.46 (0.68–8.92) | | |
| Intra-abdominal | 8 (1.7) | 6 (75) | 2 (25) | 2.87 (0.51–16.16) | | |
| Surgical site | 8 (1.7) | 6 (75) | 2 (25) | 2.87 (0.51–16.16) | | |
| **Type of antimicrobial resistance[†] (references: non-DTR)** | | | | | | |
| Non-DTR | 392 (84.3) | 346 (88.3) | 46 (11.7) | – | 19.03 | <0.001[e] |
| DTR | 73 (15.7) | 50 (68.5) | 23 (31.5) | 3.46 (1.93–6.19) | | |

*(Continued)*

**Table 5.** (Continued)

| Characteristics | n (%)b | 30-Day all-cause mortality | | UOR (95% CI) | Pearson's χ² | Pd |
|---|---|---|---|---|---|---|
| | | Survived (396) n (%)c | Died (69) n (%)c | | | |
| **Appropriateness of the antibiotic therapy†** (reference: appropriate) | | | | | | |
| Inappropriate | 205 (44.1) | 164 (80) | 41 (20) | 2.07 (1.23–3.49) | 7.72 | 0.005e |
| Appropriate | 260 (55.9) | 232 (89.2) | 28 (10.8) | – | | |

*CI*, confidence interval; *DTR*, difficult-to-treat resistant; *ICU*, intensive care unit; *SD*, standard deviation; *UOR*, unadjusted odds ratio.

aTwelve patients were excluded from the enrolled 477 patients as they were lost to follow-up.

bPercentages for the column.

cPercentages for the row.

dUnivariate analysis was conducted to test the associations between variables with mortality.

eStatistically significant (<0.05).

fIndependent t-test was conducted to test the association between Charlson Comorbidity Index with mortality.

†Retained in the final model

empiric therapy selection, reducing reliance of last-resort antibiotics, and tailoring antimicrobial treatments based on local resistance trends.

Since the DTR classification was first proposed by Kadri et al. in 2018 [8], few national studies have investigated the prevalence of DTR *P. aeruginosa* isolates. In this study, 15.3% of the isolates were classified as DTR, which is significantly higher than the prevalence reported in earlier national studies. For instance, a prospective chart review conducted between 2017 and 2020 at a tertiary healthcare institution in Beirut found that 3.1% of *P. aeruginosa* isolates were DTR [2]. Similarly, a nine-year retrospective study analyzing 1,827 *Pseudomonas* isolates from 2010 to 2018 reported a DTR prevalence of 3.33% [26]. Therefore, the higher prevalence of DTR reported in this multicenter study may reflect a growing trend of resistance in Lebanon's healthcare setting. This could be attributed to variations in IPC practices and AMS programs across different hospitals.

Regionally, DTR *P. aeruginosa* rates also show considerable variation across the Arabian Gulf countries. Between 2010 and 2021, the reported rates were 12.5% in Saudi Arabia, 19% in Kuwait, and 25.9% in Qatar [28]. Globally, data from the Antimicrobial Testing Leadership and Surveillance (ATLAS) initiative showed that DTR *P. aeruginosa* rates ranged from 6% in North America to 14.5% in Latin America, with the Middle East and Africa reporting an increase from 8% in 2018 to 10% in 2022 [9]. These findings point to an evolving pattern of AMR among *P. aeruginosa* isolates, with significant implications for treatment efficacy and healthcare costs. The variability in DTR rates between countries may be attributed to differences in demographic factors, healthcare infrastructure, access to antibiotics, and the implementation of healthcare policies.

The selection of appropriate initial antibiotic therapy is crucial in managing bacterial infections and plays a pivotal role in determining treatment success and overall patient outcomes. Empirical antibiotic therapy should be initiated promptly after obtaining diagnostic specimens, even before culture results are available. Several factors must inform the selection of empirical therapy, including current international guidelines, national treatment protocols, and updated local antibiotic resistance patterns (antibiograms) [40–42]. Additionally, patient-specific considerations—such as age, comorbidities, renal and hepatic function, immune status, pregnancy and breastfeeding status, genetic predispositions, and allergy history— are also important. Moreover, exposure-related factors, including prior colonization with MDR organisms, past infections, previous hospitalizations, and recent antibiotic use, should be accounted for to optimize the selection of initial antibiotic therapy [40–44].

In this study, nearly half of the patients received inappropriate initial antibiotic therapy, primarily due to resistance of the causative pathogens to the administered antibiotics, as determined by AST. Additionally, some patients experienced

**Table 6. Logistic regression analysis[a] of the significant predictors associated with mortality in patients with *Pseudomonas aeruginosa* infection.**

| Predictors | UOR | B | SE | Wald | AOR | 95% CI | P |
|---|---|---|---|---|---|---|---|
| **Constant** | | -3.19 | 0.36 | 74.71 | 0.04 | | < 0.001 |
| **DTR vs. non-DTR** (reference: Non-DTR) | | | | | | | |
| DTR | 3.46 | 0.90 | 0.31 | 8.11 | 2.48 | 1.32–4.63 | 0.004[b] |
| **Appropriateness of the initial antibiotic therapy** (reference: appropriate) | | | | | | | |
| Inappropriate | 2.07 | 0.15 | 0.58 | 6.95 | 1.40 | 1.04–2.35 | <0.001[b] |

*AOR*, adjusted odds ratio; *B*, coefficient for the constant in the null model; *CI*, confidence interval; *DTR*, difficult-to-treat resistance; *SE*, standard error; *UOR*, unadjusted odds ratio; *Wald*, Wald chi-square test that tests the null hypothesis.

[a]Binary logistic regression using forward stepwise analysis.

[b]Statistically significant (<0.05).

delays in receiving antibiotic treatment, with initiation occurring more than 48 hours after infection onset, while others were administered incorrect antibiotic doses. These delays raise concerns about hospital management practices and may be attributed to prolonged diagnostic processes, such as delays in obtaining culture results and AST, as well as logistical challenges including delays in antibiotic prescribing or dispensing. Such factors can adversely impact patient outcomes and contribute to poorer disease progression [40,42,45].

Several evidence-based strategies have been proven to optimize antibiotic use in clinical practice while minimizing the risk of inappropriate initial therapy. Among these, care bundles play a crucial role in guiding empirical antibiotic selection, supporting clinical decision-making, and improving prescribing practices. These bundles incorporate key elements such as novel rapid microbiological diagnostics, the use of inflammatory markers to determine the need for antibiotics, identification of patient-specific risk factors for DTR infections, and personalized antibiotic therapy, including dose optimization [46,47]. Additionally, standardized protocols such as early recognition tools and sepsis checklists are essential in ensuring timely and appropriate antibiotic administration [48]. The implementation of a multidisciplinary approach, including AMS strategies such as preauthorization of restricted antibiotics, expert consultation, and routine AMS rounds to review ongoing antibiotic therapies, has also been associated with improved antibiotic utilization and patient outcomes [49]. Future national studies are needed to evaluate the implementation of these interventions and their effectiveness in reducing the incidence of inappropriate initial therapy.

The most commonly initiated empirical antibiotics for the enrolled patients in this study were piperacillin-tazobactam and meropenem. Although carbapenems are considered a cornerstone of empirical therapy for managing serious infections caused by MDR pathogens, approximately one-third (~30%) of the *P. aeruginosa* isolates were carbapenem-resistant. There is no universally accepted threshold for local resistance rates to guide empirical antibiotic selection, but guidelines generally recommended avoiding antibiotics when local resistance rates exceed 20% [45]. Our findings suggest that carbapenem resistance among *P. aeruginosa* isolates is concerning and highlight the potential role of alternative antipseudomonal agents, such as piperacillin-tazobactam and extended-spectrum cephalosporins (e.g., ceftazidime, cefepime), which demonstrated lower resistance rates (< 16%). However, given the limitations of this observational study, further prospective studies and randomized controlled trials are needed to validate these findings and optimize empirical treatment strategies based on local resistance patterns.

The empirical use of ceftazidime-avibactam (CAZ-AVI) was observed in six patients at one of the participating hospitals, driven by high-risk clinical conditions, including ICU-acquired infections following prolonged hospital stays, urosepsis, complicated intra-abdominal infections, and ventilator-associated pneumonia. Two patients had underlying chronic kidney disease, and one had prior carbapenem therapy. While empiric CAZ-AVI use is not a routine in Lebanese hospitals, its

administration was based on clinical judgment, considering infection severity and MDR risk factors. Although the use of empirical CAZ-AVI is not recommended for the treatment of antimicrobial resistant Gram-negative infections, according to the guidelines of the Infectious Diseases Society of America (IDSA), its justification in high-risk patients with suspected MDR Gram-negative infections is acknowledged [50]. Empiric selection should be guided by local resistance patterns, patient-specific factors, prior antibiotic exposure, infection source, and illness severity. Thus, CAZ-AVI should be reserved for targeted therapy, and its empirical use should be restricted through strengthening AMS practices to optimize empirical antibiotic selection and minimize resistance development.

This study documented instances where broad-spectrum last-line antibiotics were used empirically before obtaining AST results. These antibiotics, including colistin, linezolid, and ceftaroline, are typically reserved for salvage therapy rather than routine empirical use. Upon reviewing the cases in which these antibiotics were initiated empirically, most patients presented with critical illnesses and severe infections, such as ventilator-associated pneumonia (VAP), bloodstream infections, complicated skin and soft tissue infections (cSSTIs), and infections acquired in ICU settings. Additionally, many patients developed sepsis or septic shock. While the literature strongly discourages the empirical use of last-resort antibiotics and emphasizes their role in targeted therapy, their administration may be justified in selected high-risk cases. For instance, empirical colistin therapy may be warranted in critically ill patients with invasive infections and a high likelihood of infection with XDR pathogens. Specific scenarios where empirical colistin use may be appropriate include ICU patients with VAP in settings with a high prevalence of MDR *Acinetobacter baumannii* or *P. aeruginosa*, as well as severe nosocomial infections in regions with high rates of carbapenem resistance. However, due to colistin's significant nephrotoxicity, its use should be restricted to situations where no alternative options exist [50–53]. Similarly, the IDSA recommends empirical linezolid use in cases of hospital-acquired pneumonia or VAP, cSSTIs, or bacteremia when methicillin-resistant *Staphylococcus aureus* (MRSA) is strongly suspected, especially in patients with risk factors such as nasal colonization, prior MRSA infection, recent hospitalization, or recent antibiotic exposure [54–56].

In Lebanon, the increasing prevalence of MDR pathogens—including carbapenem-resistant *A. baumannii* (CRAB), carbapenem-resistant *Klebsiella pneumoniae* (CRKP), CRPA, and MRSA—has led to a growing use of last-line antibiotics, as these pathogens are often resistant to all other available treatment options [20,25,57]. However, the overuse of these critical antibiotics poses a serious threat to AMS efforts and may contribute to further resistance development. Given these concerns, future research should assess the appropriateness of prescribing patterns for last-line antibiotics and establish clear criteria for their use. A risk-based approach should be defined to identify specific conditions and patient profiles that warrant their initiation. Additionally, restricting access to these antibiotics—requiring approval from a multidisciplinary team comprising infectious disease specialists, clinical microbiologists, and clinical pharmacists—would be a crucial step toward ensuring their judicious use and preserving their efficacy for truly resistant infections.

The study also uncovered significant concerns regarding the administration of definitive antibiotic therapy. Despite the availability of culture and AST results, a substantial number of patients experienced delays of over 24 hours in transitioning from empirical to definitive therapy. Many patients continued to receive broad-spectrum antibiotics, even when narrower-spectrum alternatives, to which the pathogens were sensitive, were readily available. Furthermore, only a small proportion of patients were transitioned to oral antibiotic therapy during their hospital stay. These findings reflect serious deficiencies in the timely optimization of antibiotic therapy, which may be attributed to several factors, including delays in reviewing and acting upon culture results to guide treatment adjustments, concerns about patient instability or infection severity, and clinician's reluctance to de-escalate due to fear of treatment failure [58]. Delays in de-escalation not only contradict AMS principles but may also indicate a tendency among physicians to prefer broad-spectrum antibiotics to avoid potential clinical deterioration. While this cautious approach may stem from concerns about ensuring rapid clinical recovery, it inadvertently contributes to the overuse of broad-spectrum agents, thereby increasing the risk of AMR development [58].

To improve antibiotic management practices, optimize patient outcomes, and mitigate the emergence of AMR, several key measures should be implemented. Clinician training programs focused on AMS principles should be established to improve adherence to de-escalation practices and promote the effective tailoring of antibiotic therapy based on AST results [43,44]. Clinicians should be encouraged to choose antibiotics with the narrowest spectrum of activity among the available sensitive options, while considering both the drug's safety profile and patient-specific factors. Promoting the timely and safe transition from IV to oral antibiotics, when clinically appropriate, is also crucial to reducing healthcare costs, shortening hospital stays, and minimizing the risk of IV-related complications [58]. Additionally, regular audits of antibiotic prescribing practices, along with the establishment and enforcement of local guidelines, will help ensure timely reviews of culture and AST results, ultimately improving antibiotic use [40–44,58].

The study found a 30-day all-cause mortality rate of 14.8%. A similar prospective study conducted between 2017 and 2020 in a Lebanese hospital among patients infected with *P. aeruginosa* reported an in-hospital mortality rate of 19.4% [2]. Mortality rates associated with *P. aeruginosa* infections vary widely across countries. They were found to be 10.5% in the United States between 2018 and 2022 [3], 15% in Switzerland between 2015 and 2021 [4], 24.2% in England between 2017 and 2019 [5], and 29.6% in Greece between 2017 and 2020 [6]. Variability in mortality rates may be influenced by differences in study design, patient populations, and healthcare system capacities. Some studies focus on specific types of *P. aeruginosa* infections, such as bloodstream infections, which tend to have higher mortality rates. Additionally, varying definitions of mortality (e.g., 14-day, 30-day, 60-day, all-cause, or attributable mortality) contribute to differences in reported rates. To improve comparability across studies, future research should prioritize the use of standardized definitions and methodologies for assessing mortality in *P. aeruginosa* infections.

This study also identified several independent predictors of mortality in patients infected with *P. aeruginosa*. Notably, infection with DTR isolates and receiving inappropriate initial antibiotic therapy were significantly associated with increased mortality risk. In particular, infection with DTR isolates was associated with approximately a 2.5-fold increase in the odds of death. This finding is consistent with previous research. For instance, a retrospective cohort study from a nationwide multicenter surveillance database in Korea found that DTR infections were associated with a higher risk of mortality [59]. The poor clinical outcomes associated with DTR *P. aeruginosa* infections are influenced by a combination of bacterial virulence, host-related factors, host-related factors, and the limitations of available antimicrobial therapy. DTR infections often occur in severely ill patients with multiple comorbidities, complicating treatment and increasing the likelihood of mortality [60].

The Kaplan-Meier survival analysis with the log-rank test did not detect a statistically significant difference in survival times between patients with DTR and non-DTR *P. aeruginosa* infections (P = 0.07). However, logistic regression identified DTR infections as a significant independent predictor of 30-day mortality (AOR = 2.48, P < 0.001). This discrepancy likely arises from fundamental differences in analytical approaches. Kaplan-Meier analysis evaluates unadjusted time-to-event distributions and does not account for confounding variables or potential violations of proportional hazards assumption. In contrast, logistic regression examines the overall association between DTR infections and mortality while adjusting for confounders, thereby isolating the independent effect of DTR infections on mortality. While Kaplan-Meier analysis focuses on survival time distributions and incorporates censored data, logistic regression simplifies the outcome into a binary variable (survival vs. death), providing a clear focus on overall mortality risk [61–63]. Thus, even if survival time distributions between DTR and non-DTR groups do not differ significantly, the logistic regression analysis demonstrates a significantly higher overall mortality risk in the DTR group. These methodological differences emphasize the importance of employing complementary statistical methods to comprehensively assess the impact of DTR *P. aeruginosa* infections on mortality outcomes.

The rapid emergence of AMR has also increased the likelihood of inappropriate initial antibiotic therapy, raising the risk of treatment failure and poor clinical outcomes [64]. In this study, patients infected with DTR *P. aeruginosa* were four times more likely to receive inappropriate initial antibiotic therapy compared to those infected with non-DTR isolates.

Furthermore, inappropriate initial antibiotic therapy was identified as an independent predictor of mortality, consistent with findings from a meta-analysis that reported a significant increase in mortality risk among patients infected with Gram-negative pathogens who received inappropriate initial antibiotic therapy [64]. In the literature, appropriate initial therapy is generally defined by the susceptibility of the prescribed antibiotic (confirmed by AST results) and its timely administration within 48 hours of infection onset [64]. A recent cohort study showed that appropriate initial therapy had a protective effect on mortality in patients with Gram-negative bloodstream infections pathogens [65].

The widespread use of broad-spectrum empirical antibiotics to treat MDR infections can inadvertently promote further resistance, perpetuating the AMR cycle. This study highlights the need for adopting recent technological advances, such as rapid diagnostic tools and artificial intelligence, to enhance the early and precise identification of pathogens and resistance genes. The implementation of rapid molecular-based tests, such as polymerase chain reaction and Matrix-Assisted Laser Desorption/Ionization–Time of Flight Mass Spectrometry, facilitates the prompt detection of MDR organisms, enabling timely, targeted antimicrobial therapy, and reducing reliance on broad-spectrum antibiotics [66]. Additionally, Next-Generation Sequencing offers a powerful approach for comprehensive resistance profiling, enabling the identification of resistance genes, novel resistance mechanisms, and whole-genome sequencing data for epidemiological surveillance [67]. Furthermore, emerging microfluidic-based diagnostic platforms offer promising solutions for rapid, real-time pathogen identification and antimicrobial resistance profiling, contributing to more precise treatment strategies. Integrating these innovations into routine clinical workflows could significantly improve antibiotic stewardship efforts, expedite targeted therapy, and minimize treatment failure and the spread of resistance. However, the widespread implementation of these advanced diagnostic tools remains limited by factors such as cost, accessibility, and laboratory infrastructure [66].

## Study limitations

This study has several limitations that should be acknowledged. First, the focus on four hospitals in Beirut may limit the generalizability of the findings, as the observed resistance patterns and antibiotic management practices might not accurately reflect the situation across hospitals in other regions of Lebanon. Consequently, these results may not provide a comprehensive picture of nationwide trends.

Second, the retrospective design and reliance on medical records for data collection present inherent limitations. Critical information that may not have been documented could have been omitted, potentially affecting the completeness of the analysis. The absence of key data restricts our ability to establish causal relationships, and the inability to account for certain confounding variables may hinder our capacity to accurately identify mortality predictors associated with DTR *P. aeruginosa* infections. For example, essential variables such as admitting diagnoses, laboratory results, follow-up cultures, clinical responses to treatment, and infection recurrence were not consistently retrieved from medical records. The lack of this information limits the depth of our analysis and introduces potential bias when evaluating factors that influence patient outcomes. Furthermore, although we accounted for comorbidities using the CCI, we were unable to include baseline severity scores (e.g., APACHE II, SOFA) due to incomplete documentation. These scores could have provided additional insights into the impact of disease severity on patient outcomes. The absence of follow-up cultures specifically restricted our ability to assess how resistance patterns may have evolved during treatment.

Third, by focusing solely on hospitalized patients who received their initial antibiotic therapy within the hospital, the study excluded outpatients, potentially leading to an overestimation of resistance patterns. As a result, the observed trends likely reflected more severe infections requiring hospitalization, limiting the applicability of the findings to less severe cases managed in outpatient settings. Fourth, the decision to include only the first infection episode for each patient excluded follow-up infections and cultures, which might have provided valuable insights into disease progression, treatment response, and the evolution of antimicrobial resistance. This approach may have introduced a bias toward simpler cases, potentially overlooking the complexities of recurrent or prolonged infections. Fifth, the exclusion of poly-microbial infections, which are frequently encountered in clinical practice, limited the study's applicability to real-world

scenarios. Polymicrobial infections often involve interactions between multiple pathogens, affecting disease severity, resistance patterns, and treatment outcomes. Excluding these cases may have overlooked important challenges in managing such infections.

Furthermore, the use of 30-day all-cause mortality as the primary endpoint likely led to an overestimation of infection-attributable deaths, as not all deaths within this timeframe could be conclusively attributed to *P. aeruginosa* infection. The retrospective design and the lack of explicit documentation in medical records further limited our ability to determine the exact causes of death. Consequently, the results should be interpreted with caution, as they may overstate the mortality burden directly associated with *P. aeruginosa* infections.

Additionally, the study only documented definitive antibiotic therapy administered during the hospital stay and did not account for therapy or outcomes following patient discharge. Due to the retrospective design, post-discharge data, including outpatient therapy and long-term outcomes, were systematically collected and were uniformly unavailable for all patients. This omission limits our ability to evaluate the influence of post-discharge antibiotic therapy on infection recovery, recurrence, and overall patient outcomes.

Finally, a key limitation of this study is the absence of molecular data, which would have helped confirm the specific mechanisms underlying carbapenem and β-lactam resistance. In particular, we did not investigate factors such as Pseudomonas-derived cephalosporinases, the regulation of OprD, or mutations in penicillin-binding proteins, all of which may contribute to the observed resistance patterns. Further molecular studies are essential to identify these mechanisms and provide a deeper understanding of the resistance profile of *P. aeruginosa* isolates in our country.

To address these limitations, future research should prioritize conducting prospective, longitudinal, multicenter cohort studies. Such studies would allow for a more comprehensive understanding of *P. aeruginosa* resistance trends across Lebanon, as well as a more detailed evaluation of antibiotic management practices. By incorporating more robust data collection methods and accounting for post-discharge therapies, these studies could provide valuable insights for health authorities and policymakers, helping them to develop targeted interventions aimed at controlling the spread of DTR *P. aeruginosa* infections and improving treatment strategies.

## 5. Conclusions

This study highlights the significant challenges posed by *P. aeruginosa* infections, particularly those caused by DTR strains, in Lebanese hospitals. The high prevalence of carbapenem and other first-line antipseudomonal resistance reflects local and regional trends in AMR, emphasizing the need for improved AMS.

The association between DTR infections and increased mortality, alongside the impact of inappropriate initial antibiotic therapy, empathize the importance of timely and appropriate treatment adjustments based on rapid diagnostics and antimicrobial susceptibility testing results. Delays in transitioning to definitive therapy and over-reliance on broad-spectrum antibiotics highlight gaps in clinical practice that require urgent attention.

To mitigate the rising threat of AMR, comprehensive IPC measures and refined empirical treatment guidelines must be prioritized. Future research should focus on expanding surveillance across Lebanon and evaluating long-term patient outcomes to guide effective interventions in controlling DTR infections and improving antimicrobial management strategies.

## Supporting Information

**S1 Table.** *In vitro* antimicrobial susceptibility testing data for *Pseudomonas aeruginosa* isolates (N=477).
(DOCX)

## Acknowledgments

None

## Author contributions

**Conceptualization:** Rania Itani.

**Data curation:** Rania Itani.

**Formal analysis:** Rania Itani.

**Investigation:** Rania Itani.

**Methodology:** Rania Itani.

**Project administration:** Rania Itani.

**Resources:** Patricia Shuhaiber, Hamza Raychouni, Carole Dib, Mariam Hassan.

**Validation:** Rania Itani.

**Visualization:** Rania Itani, Tareq L. Mukattash.

**Writing – original draft:** Rania Itani, Patricia Shuhaiber, Hamza Raychouni, Carole Dib, Mariam Hassan.

**Writing – review & editing:** Hani M. J. Khojah, Tareq L. Mukattash, Abdalla El-Lakany.

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
