## [Decision Letter · Decision Letter 0]

27 Dec 2024

PONE-D-24-46888Difficult-to-Treat Resistant Pseudomonas aeruginosa Infections in Lebanese Hospitals: Impact on Mortality and the Role of Initial Antibiotic TherapyPLOS ONE

Dear Dr. Itani,

Thank you for submitting your manuscript to PLOS ONE. After careful consideration, we feel that it has merit but does not fully meet PLOS ONE’s publication criteria as it currently stands. Therefore, we invite you to submit a revised version of the manuscript that addresses the points raised during the review process. Please submit your revised manuscript by Feb 10 2025 11:59PM. If you will need more time than this to complete your revisions, please reply to this message or contact the journal office at plosone@plos.org . Please include the following items when submitting your revised manuscript:

We look forward to receiving your revised manuscript.

Kind regards,

Giuseppe Pipitone, MD

Academic Editor

PLOS ONE

Journal Requirements:

2. In this instance it seems there may be acceptable restrictions in place that prevent the public sharing of your minimal data. However, in line with our goal of ensuring long-term data availability to all interested researchers, PLOS’ Data Policy states that authors cannot be the sole named individuals responsible for ensuring data access (http://journals.plos.org/plosone/s/data-availability#loc-acceptable-data-sharing-methods).

Reviewers' comments:

Reviewer's Responses to Questions

**Comments to the Author**

1. Is the manuscript technically sound, and do the data support the conclusions?

Reviewer #1: Partly

2. Has the statistical analysis been performed appropriately and rigorously? 

Reviewer #1: Yes

3. Have the authors made all data underlying the findings in their manuscript fully available?

Reviewer #1: No

4. Is the manuscript presented in an intelligible fashion and written in standard English?

Reviewer #1: Yes

5. Review Comments to the Author

Reviewer #1: The authors submitted a manuscript investigating Pseudomonas aeruginosa infections in 4 Lebanese tertiary care hospitals, with a particular focus on the prevalence and impact of DTR isolates, as well as the apppropriateness of initial antibiotic therapy and predictors of 30-day in-hospital mortality.

- The manuscript describes exclusion criteria (e.g., <48 h hospital stay, incomplete records, co-infections). This is raises the risk of selection bias. please expand on the rationale for excluding patients discharged or leaving the hospital within 48 hours (this will be further touched upon later in my report). Were some of those patients potentially stable, or were they left out simply because they might not have had time to develop or treat an infection? The flow diagram (Figure 1) is helpful, but more detail on potential biases from these exclusions would be beneficial.

- This endpoint can overestimate infection-attributable deaths. The authors acknowledge this limitation. discuss how infection-related mortality was confirmed, or whether chart reviews attempted to differentiate conflicting causes of death.

a. A Kaplan-Meier with log-rank comparisons for time-to-mortality is used. The difference for time-to-event between DTR vs. non-DTR just misses significance (p=0.07), but the difference based on appropriateness of initial therapy is significant (p=0.03). The text reports “mean survival time” and 95% CIs that appear somewhat narrow (e.g., for the DTR group in Table 4: 4.06–7.93 days?). The survival curves appear to go out to 30 days; thus, the meaning of that confidence interval needs more clarity. Are these the 95% CIs for median survival times?

b. The approach to univariate → multivariable logistic regression is described, with a p < 0.2 cut-off for inclusion. The text says that forward stepwise selection was used, and that variance inflation factors (VIF > 10) triggered exclusion. This is reasonable, but it is unclear which variables were excluded due to collinearity, I will mention this again below in detailed analysis. More detail would help: for instance, was the site of infection acquisition (ICU vs. non-ICU) collinear with nosocomial vs. community infection? Also, the final model includes female sex, DTR infection, and appropriateness of therapy. Yet the univariate table also identified age ≥ 65, CCI, and glucocorticoid use as having p<0.2. Possibly these were excluded for collinearity reasons, but clarifying this in the text would be important to ensure transparency.

c. The mortality rate of 14.8% fits within expected estimates. The discussion section offers does not offer sufficient rationale or thoughts as to why female patients had higher mortality. Assuming that any difference in comorbidity that is gender based would be accounted for in a logistic regression model; please expand on potential epidemiologic reasons (e.g., healthcare-access?) . Again this will be mentioned in more detail below

d. Despite the non-significant p=0.07 in the survival analysis, the logistic regression finds DTR to be an independent predictor of mortality (AOR 3.11). This discrepancy likely arises from how survival times were partitioned and the additional effect of confounders in the multivariable model. Please provide provide a short note about this distinction in the Discussion. Some reasons may require interpretation as to why the log-rank p-value for time-to-mortality differs from the logistic regression results.

e. More detail on how the authors handled or imputed missing data (if any was imputed at all) is needed. As stated a major data gap is post-discharge therapy and outcomes which is understandable given the study’s retrospective design, but the authors might mention whether they attempted to collect data from outside the hospital within that context for some patients or if that was uniformly unavailable.

-Did all four hospitals use the same AST platforms and interpretative standards (CLSI vs. EUCAST)?

- Table 1 indicates that 6 patients received CAZ-AVI empirically. Please provide context for this empiric practice and whether the AST associated with these 6 patients justified appropriate CAZ-AVI use.

-The same can be said about the empiric usage of several agents such as ceftaroline, linezolid, and colistin. Please elaborate on whether these are errors. By empiric, most readers will consider this to represent Pre-AST.

Given that data from your region is harder to come by your cohort can reveal more and inform more about the epidemiology:

- Please inform readers of the routine AST agents performed for pseudomonas aeruginosa. Is CAZ-AVI susceptibility testing reflex tested with carbapenem non-susceptibility?

-What percentage of isolates were resistant to CAZ-AVI?

-Is Ceftolozane-tazobactam available in any of the medical centers included in the study? If yes, what % os isolates were non-susceptible to TOL-TAZ?

-Is Cefiderocol available in any of the medical centers included in the study? Was resistance encountered?

-Did you encounter on follow up cultures resistance to any B-lactam derived agents including cephalosporins or carbapenems or CAZ/AVI during treatment

-For definitive antibiotic therapy, also seen in Table 2; I am assuming Fosfomycin is the IV formulation. Please indicate (IV) in brackets in the table. Many readers are only familiar with PO FOS in the setting of UTIs.

You mention the absence of national surveillance data and “the paucity of well-designed studies” on DTR PA in Lebanon. Are there any recent steps taken nationally to track AMR? A few lines referencing the challenges of implementing surveillance is helpful in epidemiologic studies.

- With regards to the 4 hospitals in Beirut city, do they differ in patient populations, bed capacity, or infection-control measures?

- Were patients <48 hrs excluded because they might not have had time to be fully evaluated or treated? Other reasons?

-Please quantify the effect of excluding from your study patients who left the hospital within 48 hours of acquiring infection. How does this exclusion impact the observed results and conclusions. A bias may have been inadvertently introduced by excluding mild infections that had resolved more rapidly. This is relevant to the observed significant findings. How might this exclusion have impacted the stepwise model, and ultimately across significant subgroups as it pertains to primary outcomes.

-Did any variables meet the >10 VIF threshold and thus get excluded from the final model? If so, which ones?

-Including a short table of final retained variables following univariate analysis would help or otherwise incorporating, bolding, or annotating with symbols may be sufficient to show the final included variable at a glance.

-How might confounding variables may shift results showing survival analysis results demonstratinhg a near-significant difference in log-rank tests for DTR vs. non-DTR (p=0.07), especially given that logistic regression finds DTR to be a significant independent predictor.

-Given the relatively advanced age and high comorbidity burden, do you have severity-of-illness scores (e.g., APACHE II, SOFA on admission)? You mention a mean SOFA of ~5.89 for those who developed sepsis, but it might be helpful to note baseline severity if available, to interpret outcomes.

-Did you observe any association between specific infection sites and higher rates of DTR or carbapenem resistance?

-For patients whose antibiotics were inappropriate due to resistance, do you have data on local antibiotic guidelines that might reduce these mismatches?

-noting that 30 patients experienced >48 hours delay in therapy initiation. Was this due to diagnostic delays, or were there other logistical/administrative reasons?

-Similar to above, what are the major prevailing reasons for delays in de-escalation or switching after the results of AST became available (noting that this occurred in almost 19%)

-Going over the multivariable analysis in more detail and table 6 showing three independent mortality predictors:

1. Female sex (AOR=4.41)

2. DTR infection (AOR=3.11)

3. Inappropriate initial therapy (AOR=1.43)

- Did you stratify or examine the distribution of comorbidities (hypertension, CHF, etc.) by sex? Could confounders remain unrecognized (e.g., older women with multiple comorbidities)?

- Could ICU admission or other severity metrics modify (mediate or confound) the effect of sex on mortality?

Please touch on these topics to increase the value of the discussion

- To what extent do isolates within your cohort demonstrate simultaneous carbapenem resistance with susceptibility to cefepime or Pip/Taz? Briefly touch on the potential mechanisms of carbapenem resistance in isolates with concomitant susceptibility to 4th gen cephalosporins or Pip/taz. Discuss whether this particular pattern is unique to your epidemiologic setting or healthcare system. This is especially relevant since you have suggested a change in empiric practices based on this data. Does this pattern of pseudomonas aeruginosa resistome generalize to other institution in your country or region? Is there any molecular regional data pertaining to oprD downregulation that you can allude to?

- I would recommend against suggesting practice changes based on observational studies in a manuscript. The more appropriate reflection based on the limitations of this study design is somewhere along the lines of ‘interesting results but need randomized data to inform practice’ etc.

- another limitation worth mentioning along with the lack of any molecular data within this study or mention of any specific PDCs, outer porin regulation, and PBP mutations among others.

- Would any rapid diagnostic tools expedite targeted therapy in the described context of a lack of standardized antibiotic protocols and the necessity of improved stewardship?

- Within the framework of ther results showing that inappropriate initial therapy (including timing/dose errors) being common; haave you considered analyzing whether a “bundle approach” (i.e., checklists for sepsis, analysis of DTR risk factors/daily stewardship rounds) could systematically lower this rate?

- Please ascertain within your cohort the instances of inappropriate antibiotic therapy (with Pip/Taz or cefepime) where a carbapenem should have been prescribed empirically (based on risk factors).

Suggest emphasizing in the discussion any missed covariates (e.g., baseline severity scores, or comorbid conditions) that might influence results.

-AST data from Lebanon would be interesting to a global audience. If you are able to provide raw AST data on all isolates in a supplement, it would be appreciated and carry value.

6. PLOS authors have the option to publish the peer review history of their article (what does this mean? ). If published, this will include your full peer review and any attached files.

**Do you want your identity to be public for this peer review?** For information about this choice, including consent withdrawal, please see our Privacy Policy .

Reviewer #1: **Yes: ** Mohamad Yasmin

---

## [Author Response · Author response to Decision Letter 0]

22 Feb 2025

Dear Prof. Giuseppe Pipitone, MD

Academic Editor

PLOS ONE

PONE-D-24-46888

Thank you for giving us the opportunity to submit a revised draft of our manuscript titled “Difficult-to-Treat Resistant Pseudomonas aeruginosa Infections in Lebanese Hospitals: Impact on Mortality and the Role of Initial Antibiotic Therapy” to PLOS ONE. We appreciate the time ‎and effort that you and the reviewer have dedicated to providing valuable feedback on our ‎manuscript. We are grateful to your insightful comments. We have been able to ‎incorporate changes to reflect all of the suggestions you and the reviewer have provided. Those changes are ‎highlighted within the manuscript. ‎

Please see below, in blue, for point-by-point responses to the comments and ‎concerns.

Reviewers' comments:

Reviewer's Responses to Questions

Comments to the Author

1. Is the manuscript technically sound, and do the data support the conclusions?

Reviewer #1: Partly

Authors’ response:

Dear Reviewer. Thank you for taking the time to extensively review our manuscript and for providing valuable comments and suggestions. We sincerely appreciate your insightful feedback, which has significantly contributed to improving the quality and clarity of our work. We have thoroughly reviewed each of your comments and made the appropriate revisions. Below, you will find our detailed responses to your suggestions, along with the changes implemented in the manuscript. We hope these revisions meet your expectations and that the manuscript is now suitable for publication.

2. Has the statistical analysis been performed appropriately and rigorously?

Reviewer #1: Yes

3. Have the authors made all data underlying the findings in their manuscript fully available?

Reviewer #1: No

Authors’ response:

We fully understand the importance of making underlying data available. However, the datasets generated and analyzed during this study cannot be publicly available due to restrictions imposed by the Institutional Review Boards (IRBs) and ethical committees of the participating hospitals. These restrictions are in place to protect patient privacy and confidentiality, as the data contain potentially identifying or sensitive patient information. Public sharing of this data could risk patient re-identification. Additionally, given that this study assessed hospital management practices, further confidentiality concerns apply. In accordance with ethical standards, access to these datasets is possible only upon formal request to the corresponding author (r.itani@bau.edu.lb) or to the IRB at Beirut Arab University (irb@bau.edu.lb), accompanied by approval from the hospitals' IRBs and ethical committees. All relevant data supporting the conclusions of this study have been included within the article. Furthermore, the data will be securely stored in Beirut Arab University’s institutional repository, ensuring long-term availability and accessibility through the IRB. We have incorporated these details into the Data Availability Statement in the manuscript.

4. Is the manuscript presented in an intelligible fashion and written in standard English?

Reviewer #1: Yes

5. Review Comments to the Author

Reviewer #1: The authors submitted a manuscript investigating Pseudomonas aeruginosa infections in 4 Lebanese tertiary care hospitals, with a particular focus on the prevalence and impact of DTR isolates, as well as the appropriateness of initial antibiotic therapy and predictors of 30-day in-hospital mortality.

- The manuscript describes exclusion criteria (e.g., <48 h hospital stay, incomplete records, co-infections). This is raises the risk of selection bias. please expand on the rationale for excluding patients discharged or leaving the hospital within 48 hours (this will be further touched upon later in my report). Were some of those patients potentially stable, or were they left out simply because they might not have had time to develop or treat an infection? The flow diagram (Figure 1) is helpful, but more detail on potential biases from these exclusions would be beneficial.

Authors’ response:

Thank you for your thoughtful comment and for highlighting the potential risk of selection bias introduced by the exclusion criteria. We appreciate the opportunity to clarify the rationale for excluding patients discharged or leaving the hospital within 48 hours, as well as the potential biases associated with this decision. Patients discharged or who left the hospital within 48 hours of infection acquisition were excluded if they did not receive their initial antibiotic therapy within the hospital. Since our study focuses on assessing the appropriateness of initial antibiotic therapy administered during hospitalization, including these patients would have limited our ability to evaluate this aspect consistently. Additionally, discharge medications were not systematically documented in this study, further limiting our ability to assess post-discharge antibiotic use and its influence on treatment outcomes. We have handled these issues as follows:

2. Methods Section

2.2. Study population

“The study included hospitalized adult patients aged 18 years and older with a confirmed diagnosis of P. aeruginosa infection who received treatment at any of the participating hospitals during the study period. For patients with recurrent P. aeruginosa infections, only the first infection episode was documented to avoid overrepresentation of individual patients in the dataset and to reflect baseline antimicrobial resistance profiles without the confounding influence of prior treatments or recurrent infections.

The exclusion criteria were as follows: cases of colonization rather than active infection, polymicrobial infections or co-infections, patients treated as outpatients (i.e., those not admitted to the hospital), subsequent hospitalizations due to recurrent infections, patients discharged or who left the hospital before receiving initial antibiotic therapy, and incomplete medical records lacking antimicrobial susceptibility testing (AST) results or other crucial medical information. These criteria were designed to establish a focused and representative study population of hospitalized patients with active P. aeruginosa infections. By excluding cases involving recurrent infections, co-infections, or colonization, the study aimed to minimize confounding factors that could obscure the specific impact of P. aeruginosa on clinical outcomes. This approach enabled a targeted evaluation of the appropriateness of initial antibiotic therapy for infections caused by P. aeruginosa and facilitated the estimation of the associated in-hospital mortality.”

Study limitations (3rd paragraph)

“Third, by focusing solely on hospitalized patients who received their initial antibiotic therapy within the hospital, the study excluded outpatients, potentially leading to an overestimation of resistance patterns. As a result, the observed trends likely reflected more severe infections requiring hospitalization, limiting the applicability of the findings to less severe cases managed in outpatient settings. Fourth, the decision to include only the first infection episode for each patient excluded follow-up infections and cultures, which might have provided valuable insights into disease progression, treatment response, and the evolution of antimicrobial resistance. This approach may have introduced a bias toward simpler cases, potentially overlooking the complexities of recurrent or prolonged infections. Fifth, the exclusion of polymicrobial infections, which are frequently encountered in clinical practice, limited the study's applicability to real-world scenarios. Polymicrobial infections often involve interactions between multiple pathogens, affecting disease severity, resistance patterns, and treatment outcomes. Excluding these cases may have overlooked important challenges in managing such infections.”

We also revised Figure 1 to provide additional detail on the exclusion of patients discharged without receiving initial antibiotic therapy during their hospital stay.

- This endpoint can overestimate infection-attributable deaths. The authors acknowledge this limitation. discuss how infection-related mortality was confirmed, or whether chart reviews attempted to differentiate conflicting causes of death.

Authors’ Response:

Thank you for your insightful comment. We agree that the use of 30-day all-cause mortality as the primary endpoint may overestimate infection-attributable deaths, as not all deaths within this timeframe can be linked to P. aeruginosa infection. This limitation is inherent to our study's retrospective design and the reliance on medical records, which often lack explicit documentation of the exact cause of death.

In our chart reviews, we aimed to identify potential contributors to mortality; however, detailed clinician-assigned cause-of-death information was not consistently available. This limitation restricted our ability to differentiate infection-related deaths from those caused by underlying comorbidities, other infections, or non-infectious conditions. We have acknowledged this challenge in the Study limitations section (4th paragraph), as follows:

“Furthermore, the use of 30-day all-cause mortality as the primary endpoint likely led to an overestimation of infection-attributable deaths, as not all deaths within this timeframe could be conclusively attributed to P. aeruginosa infection. The retrospective design and the lack of explicit documentation in medical records further limited our ability to determine the exact causes of death. Consequently, the results should be interpreted with caution, as they may overstate the mortality burden directly associated with P. aeruginosa infections.”

a. A Kaplan-Meier with log-rank comparisons for time-to-mortality is used. The difference for time-to-event between DTR vs. non-DTR just misses significance (p=0.07), but the difference based on appropriateness of initial therapy is significant (p=0.03). The text reports “mean survival time” and 95% CIs that appear somewhat narrow (e.g., for the DTR group in Table 4: 4.06–7.93 days?). The survival curves appear to go out to 30 days; thus, the meaning of that confidence interval needs more clarity. Are these the 95% CIs for median survival times?

Authors’ response:

Thank you for your valuable observation. You are correct that the 95% confidence intervals (CIs) reported for survival times in Table 3 and Table 4 were originally for the median survival time rather than the mean. We acknowledge that this distinction could lead to confusion for readers, especially since Kaplan-Meier curves typically extend to 30 days while the reported CIs for the DTR group were relatively narrow (e.g., 4.06–7.93 days). To address this, we have revised Table 3 and Table 4 to now report the mean survival time and its 95% CI, as this provides a more straightforward and interpretable metric in this context.

b. The approach to univariate → multivariable logistic regression is described, with a p < 0.2 cut-off for inclusion. The text says that forward stepwise selection was used, and that variance inflation factors (VIF > 10) triggered exclusion. This is reasonable, but it is unclear which variables were excluded due to collinearity, I will mention this again below in detailed analysis. More detail would help: for instance, was the site of infection acquisition (ICU vs. non-ICU) collinear with nosocomial vs. community infection? Also, the final model includes female sex, DTR infection, and appropriateness of therapy. Yet the univariate table also identified age ≥ 65, CCI, and glucocorticoid use as having p<0.2. Possibly these were excluded for collinearity reasons, but clarifying this in the text would be important to ensure transparency.

Authors’ response:

Thank you for your valuable comment and for highlighting the need for additional clarity regarding variable inclusion and exclusion in the multivariable logistic regression model. We have revised the methodology section as follows:

2.6. Statistical analysis (2nd, 3rd, and 4th paragraphs)

“Univariate analysis was conducted to assess associations between independent variables and 30-day all-cause mortality, utilizing Pearson's chi-square (χ²) test for categorical variables and independent t-tests for continuous variables. Variables with a P-value < 0.2 in the univariate analysis were included in the multivariable logistic regression model. The variables initially included were: age group (≥ 65 years vs. 18–64 years), sex (female vs. male), site of infection acquisition (community-acquired vs. nosocomial infection), hospital department of infection acquisition (ICU vs. non-ICU), coronary artery disease, congestive heart failure, CCI, previous glucocorticoid use within the last four weeks, type of antimicrobial resistance (DTR vs. non-DTR P. aeruginosa infection), and appropriateness of the initial antibiotic therapy (appropriate vs. inappropriate).

Multicollinearity was assessed using variance inflation factors (VIF), with a threshold of 10 for exclusion. All VIF values were below 2.5, indicating no significant multicollinearity among the predictors. Conceptually related variables were carefully reviewed for inclusion based on clinical relevance and contribution to model fit. After careful consideration, ICU vs. non-ICU was retained for its stronger relevance to patient outcomes, while nosocomial vs. community-acquired infection was excluded. Similarly, congestive heart failure and coronary artery disease were excluded due to their conceptual overlap with CCI, which was retained as a more comprehensive measure of comorbidity burden.

The final multivariable logistic regression model was developed using a stepwise forward selection approach, which sequentially added variables based on their statistical contribution to the model. Confounding variables were retained in the model if their inclusion altered the coefficient of any significant variable by 10% or more. The glucocorticoid use, although identified in the univariate analysis (P < 0.2), was excluded during forward selection as it did not significantly contribute to the final model after accounting for other variables. The goodness-of-fit for the model was assessed using Hosmer-Lemeshow test. Adjusted odds ratios (AOR) with 95% confidence intervals and P-values were calculated for all variables in the model.”

c. The mortality rate of 14.8% fits within expected estimates. The discussion section offers does not offer sufficient rationale or thoughts as to why female patients had higher mortality. Assuming that any difference in comorbidity that is gender based would be accounted for in a logistic regression model; please expand on potential epidemiologic reasons (e.g., healthcare-acce

---

## [Editor Report · Decision Letter 1]

14 Mar 2025

Difficult-to-Treat Resistant Pseudomonas aeruginosa Infections in Lebanese Hospitals: Impact on Mortality and the Role of Initial Antibiotic Therapy

PONE-D-24-46888R1

Dear Dr. Rania Itani,

We’re pleased to inform you that your manuscript has been judged scientifically suitable for publication and will be formally accepted for publication once it meets all outstanding technical requirements.

Kind regards,

Giuseppe Pipitone, M.D.

Academic Editor

PLOS ONE
---

## [Editor Report · Acceptance letter]

PONE-D-24-46888R1

PLOS ONE

Dear Dr. Itani,

I'm pleased to inform you that your manuscript has been deemed suitable for publication in PLOS ONE. Congratulations! Your manuscript is now being handed over to our production team.

Kind regards,

on behalf of

Dr. Giuseppe Pipitone

Academic Editor

PLOS ONE